# Preparation and Properties of Flame Retardant and Antistatic Foamed Wood–Plastic Composite with APP/ZB System

Zhitao Lei [1,2,†], Jie Liu [1,2,†], Yating Zhao [1,2,†], Xuesong Zhao [3,*] and Qi Li [1,2,*]

[1] College of Material Science and Art Design, Inner Mongolia Agricultural University, Hohhot 010010, China; leilanzt@163.com (Z.L.); 13193770827@163.com (J.L.); m18847164193@163.com (Y.Z.)
[2] Inner Mongolia Key Laboratory of Sandy Shrubs Fibrosis and Energy Development and Utilization, Hohhot 010018, China
[3] College of Mechanical and Electrical Engineering, Inner Mongolia Agricultural University, Hohhot 010010, China
* Correspondence: zhao_xuesong555@163.com (X.Z.); li_qi555@163.com (Q.L.)
† These authors contributed equally to this work.

**Abstract:** With the aggravation of fire and smoke pollution, it is urgent to develop green, lower-cost and high-performance Foamed Wood–Plastic Composite (FWPC) to meet the standards of antistatic and flame retardant in practical application. Therefore, the flame retardant and antistatic FWPCs were prepared by compression molding in this study. High-density polyethylene (PE-HD) and Salix wood flour were used as main raw materials, and azodicarbonamide (AC) was used as foaming agent; Nano-carbon black (Nano-CB) was used as antistatic filler, and ammonium polyphosphate (APP) and zinc borate (ZB) were used as flame retardants. The static bending strength and elastic modulus of FWPC-20 were up to 30.01 MPa and 2636 MPa, respectively, which can meet the commercial application of wood–plastic decorative board. The logarithm of surface resistivity and volume resistivity of FWPC-20 was kept at eight, indicating that it has antistatic effect. The residual carbon rate of FWPC-20 increased to 38.58% at 800 °C, indicating that FWPC had high thermal stability. The minimum heat release rate of FWPC-20 was 226.75 kw/m$^2$, and the average heat release rate was 110.53 kw/m$^2$. The total heat release was 66.96 MJ/m$^2$, and the Limit Oxygen Index was 27.3%, which indicated that FWPC-20 had flame retardant and smoke suppression effects. This study provides a low-cost and simple method for the design of flame retardant, antistatic and high-performance FWPC, and has broad application prospects in the fields of packaging and construction.

**Keywords:** Salix; high-density polyethylene; foamed wood–plastic composites; antistatic; flame retardant

## 1. Introduction

The density and raw material cost of the biomass wood–plastic composite material can be reduced by adding a foaming agent during the hot-pressing process. At the same time, foaming can also imprint a unique texture to the surface of the product, improving its sound and heat insulation properties. Foamed wood–plastic composites were based on wood flour and plastic. The performance of the composite material can be improved by adding a foaming agent during the hot-pressing process. Gas was released by decomposition under the action of high temperature of foaming agent. Foamed wood–plastic composites were formed by introducing tiny air bubbles into composites through air nuclei [1,2]. However, wood powder and plastic were flammable. When they burn, the flame temperature is high and spreads fast. At the same time, a large amount of smoke and dust, poisonous and harmful gases are released [3,4]. Therefore, preparation of flame-retardant wood–plastic composites with excellent performance has become a research hotspot.

In recent years, related wood–plastic composites have been extensively studied by researchers. Zhang et al. [5] prepared a biochar/wood–plastic composite by extrusion

molding and tested its mechanical properties and flame retardancy. The results showed that with the increase in biochar content, the mechanical properties of wood–plastic composites increased first and then decreased. The addition of $Mg(OH)_2$ and $Al(OH)_3$ can significantly improve the flame-retardant properties of the composites. When of $Mg(OH)_2$ is added in the amount of 40 wt%, the flame-retardant effect of the composites is the best.

The effect of red pottery content on the properties of high-density polyethylene wood–plastic composites was studied by Li et al. [6]. The results showed that when the content of red clay was 5%, the mechanical properties of wood–plastic composites were the best. At this time, flame retardancy of the composite was obviously improved by red clay. Polypropylene (PP)/organic montmorillonite (OMMT) composites were prepared by injection molding by Chow [7]. The effect of the amount of antistatic agent (3–9 wt%) on its performance was studied. XRD results show that antistatic agent can improve the intercalation of PP/OMMT composites and affect the hardness, crystallinity and resistivity of the materials. The thermal decomposition temperature of the material was increased. Yu et al. prepared flame-retardant and antistatic wood–plastic composites with carbon black (CB), ammonium polyphosphate (APP) and expandable graphite (EG) as modifiers. The results show that the antistatic effect of the composites was obviously improved by carbon black and EG. Cone calorimeter test results showed that the smoke emission, total smoke emission (TSR) and carbon monoxide production of wood–plastic composites were reduced. In the combustion process, the heat output from the material was reduced, and the carbon residue rate was increased due to the synergistic effect of EG and APP [8].

High-efficient flame-retardant ammonium polyphosphate (APP) with molecular formula $(NH4)_n + 2P_nO_{3n+1}$ is a low-cost, nontoxic and environmentally friendly inorganic phosphorus flame-retardant compound. Gaseous ammonia is released during combustion. Therefore, in the combustion process, the concentrations of oxygen and other gases were decreased by the increase in ammonia volume. In the condensation stage, APP was decomposed and poly phosphoric acid was produced. Dehydration and carbonization of polymers were promoted. Oxygen can be isolated by a carbon layer formed by a flame retardant [9,10]. One of the first non-halogen flame retardants, zinc borate (ZB), has the chemical formula $2ZnO·3B_2O_3·3.5H_2O$. ZB has excellent flame-retardant and smoke-suppression properties. It is non-toxic, pollution-free and easy to popularize. Therefore, it is often used to enhance the flame-retardant properties of wires and cables, anti-rust coatings, plastic and rubber [11,12].

The decomposition temperature of azodicarbonamide (AC) can be reduced by zinc oxide (ZnO). As a result, the foaming rate of AC is improved, and the foaming effect is better. The addition of Nano-$CaCO_3$ can increase the number of foaming sites and bubble of WPC and make the bubble shape more uniform. Therefore, foaming agent (AC), initiator (ZnO) and nucleating agent (Nano-$CaCO_3$) were added to the Salix/PE-HD composites. The foaming process can effectively reduce their density and weight. At the same time, cellular structure can prevent crack propagation and improve the problem of poor toughness of materials. In addition, nano conductive carbon black (Nano-CB) and synergistic flame-retardant APP/ZB were added to FWPC system to prepare flame-retardant and antistatic Salix/PE-HD foamed composites. After APP/ZB was added to FWPC, it displayed better flame-retardant and smoke-suppression performance on the basis of maintaining its good mechanical properties. At the same time, its thermal stability was improved. Therefore, this study introduces a simple method to develop flame-retardant, antistatic and high-performance FWPC which has broad application prospects in the multifunctional utilization of wood and packaging, transportation and decoration.

## 2. Materials and Methods

### 2.1. Materials

The high-density polyethylene (PE-HD) was supplied by South Korea LG, Zhejiang, China. Salix wood flour (particle size is 80–100 mesh) was supplied by Man Lai Township Ordos City Inner Mongolia, China (it has both sapwood and heartwood). The age of

Salix is 5–6 years. The moisture content is 2 wt%. Coupling agent (KH550) was provided by Nanjing Chuang Shi Chemical Auxiliary Co., Ltd., Nanjing, China. Stearic acid was supplied by Xilong Science Co., Ltd., Shantou, China. Ethanol (95 wt%) was supplied by Tianjin Feng Chuan Chemical Reagent Technology Development Co., Ltd., Tianjin, China. Azodicarbonamide (AC) was supplied by Jiangsu Thorpe Co., Ltd., Jiangsu, China. NaOH was supplied by Tianjin Hui hang Chemical Technology Co., Ltd., Tianjin, China. Zinc oxide (ZnO) was supplied by Hebei Bai YI LIAN Chemical Technology Co., Ltd., Baoding, China. Nano-calcium carbonate (Nano-$CaCO_3$) was supplied by Shanghai Yuanjiang Chemical Industry and Foreign Trade Department, China. Nano-carbon black (Nano-CB, GM-TY5) was purchased from Changzhou Heng feng Nanotechnology Co., Ltd., Changzhou, China. Ammonium polyphosphate (APP-105, surface modified by epoxy resin, phosphorus content ≥29 wt%, nitrogen content ≥14 wt%, particle size less than 15 μm) was supplied by Shenzhen Jing Cai Chemical Co., Ltd., Shenzhen, China. Zinc borate (3.5 water zinc borate) was supplied by Lin Yi Heng Shuo Chemical Professional Flame Retardant Co., Ltd., Lin Yi, China.

### 2.2. Preparation of Wood–Plastic Composites

The design density of FWPC was 0.9 g/cm$^3$. The size of FWPC was designed to be 15 mm × 15 mm × 4 mm. The total mass of FWPC was the sum of PE-HD, Salix powder, Nano-CB and APP/ZB. The mass ratio of Salix wood powder and PE-HD was 4:6. As shown in Table 1, the fixed amount of coupling agent (KH550) was 3 wt%, the fixed amount of stearic acid was 0.6 wt%, the fixed amount of AC was 1.5 wt%, the fixed amount of Nano-$CaCO_3$ was 3 wt% and the fixed amount of Nano-CB was 8 wt% (wt.% was the percentage of the total mass of FWPC). The fixed addition amount of zinc oxide was 20 wt% of the mass of foaming agent (AC). Composite ammonium polyphosphate (APP) and zinc borate (ZB) were added in proportion (4:1), and the amounts of APP/ZB synergistic flame retardant were 0,5 wt%, 10 wt%, 15 wt% and 20 wt%, respectively. The obtained FWPCs were named FWPC-0, FWPC-5, FWPC-10, FWPC-15, FWPC-20, respectively (wt.% was the percentage of the total mass of FWPC. The following values of FWPC represent the amount of APP/ZB added).

**Table 1.** Formula of Flame-Retardant and Antistatic Foamed Wood–Plastic Composite.

| Sample | APP/ZB | KH550 | Stearic Acid | AC | Nano-$CaCO_3$ | Nano-CB |
|---|---|---|---|---|---|---|
| FWPC-0 | 0 | 3 | 0.6 | 1.5 | 3 | 8 |
| FWPC-5 | 5 | 3 | 0.6 | 1.5 | 3 | 8 |
| FWPC-10 | 10 | 3 | 0.6 | 1.5 | 3 | 8 |
| FWPC-15 | 15 | 3 | 0.6 | 1.5 | 3 | 8 |
| FWPC-20 | 20 | 3 | 0.6 | 1.5 | 3 | 8 |

Firstly, the Salix powder was dried and modified with 12 wt% NaOH solution at 60 °C for 60 min, and then washed with distilled water to neutral. KH550 was hydrolyzed with 95 wt% ethanol, mixed with modified wood flour and Nano-CB, respectively, and dried for later use. The temperature of the double roller of the mixer was raised to 160 °C and 170 °C, and PE-HD, related foaming additives, modified wood powder, modified Nano-CB, and flame retardant were added in turn for uniform mixing. Finally, the mixture was chopped and put into the mold, which was formed by hot pressing and cold pressing. The hot-pressing temperature, pressure and time were 190 °C, 5 MPa and 7 min, respectively. The cold-pressing pressure and time were 7 MPa and 10 min.

### 2.3. Characterization

The static bending strength and elastic modulus were tested by Electronic universal testing machine (Model WDW-20A, Jinan Tianchen Machinery Manufacturing Co., Ltd., Jinan, China). The tensile strength was tested by a Mechanical testing machine (Model AG-IC, Shimadzu Instruments (Suzhou) Co., Ltd., Suzhou, China). The impact strength

was tested by a Pendulum Impact Tester (Model ZBC7151-B, Meters Industrial Systems Co., Ltd., Shenzhen, China). Dynamic mechanical properties of the composites were measured by using a dynamic mechanical analyser (Model Q800, TA company, Floor Boston, MA, USA) under Multi-Frequency-Strain mode. Limit Oxygen Index of the samples was measured by Oxygen Index Meter (Model JF-3, Nanjing Jiangning District Analysis Instrument Factory, Nanjing, China). The electrical conductivity of FWPC was tested by a resistivity tester (Model BEST-212, Beijing Beiguang Precision Instrument Equipment Co., Ltd., Ningbo, China). Thermal degradation of the composites was studied by a thermogravimetric analyzer, (STA409PC, NETZSCH, Netzsh, Germany). The reaction-to-fire of FWPC was tested by (FTT) cone calorimeter (Model MOTS, Motis Company, Suzhou, China). Microstructure images of samples were obtained by scanning electron microscopy (Phenom Pro, Funa Scientific Instruments (Shanghai) Co., Ltd., Amsterdam, The Netherlands).

## 3. Results

### 3.1. Mechanical Properties

The trend of the mechanical properties of FWPC is shown in Figure 1.

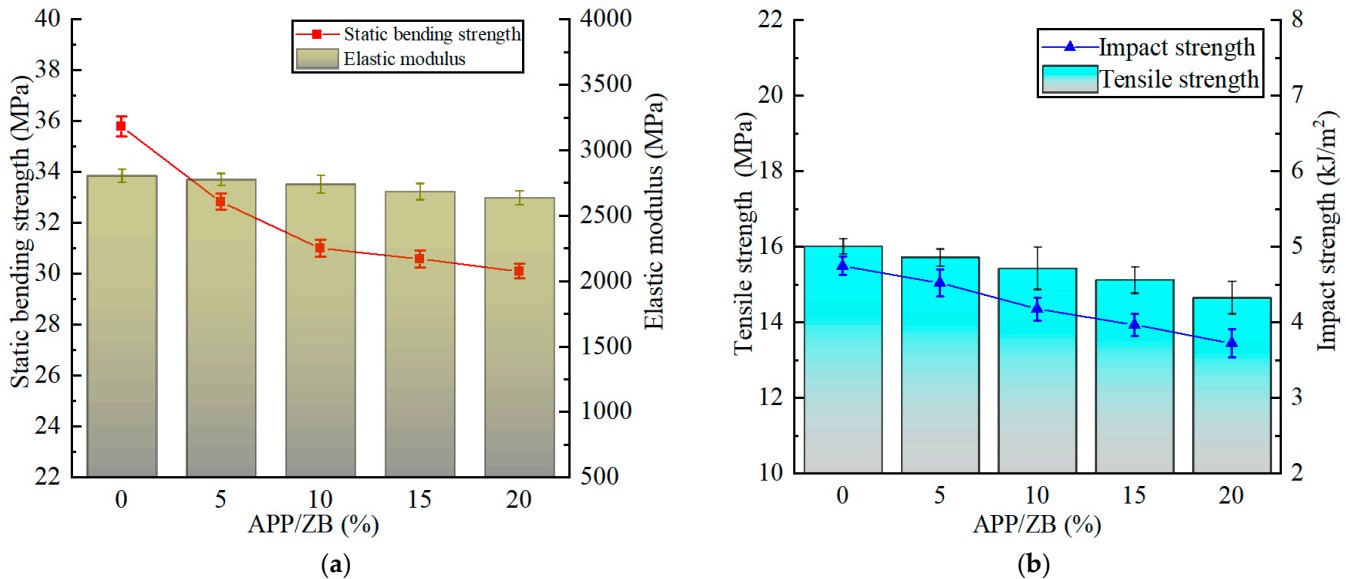

**Figure 1.** Effect of different APP/ZB content on static bending strength and elastic modulus (**a**). Effect of different APP/ZB content on tensile strength and impact strength (**b**).

The static bending strength, elastic modulus, tensile strength and impact strength of FWPC decreased while the content of APP/ZB synergistic flame retardant increased. When 20 wt% flame-retardant APP/ZB was added to the composites, the mechanical properties of FWPC-20 were 30.01 MPa, 2636 MPa, 14.65 MPa and 3.72 kJ/m$^2$, respectively. Compared with FWPC-0, the retention rates of mechanical properties of FWPC-20 were 84.1%, 94.0%, 91.5% and 78.3%, respectively. This is because the polarity of APP/ZB and PE-HD matrix were different, which led to their poor compatibility. With the gradual increase in APP/ZB content, the dispersion of a large number of powder particles in the composite system is weakened. The agglomeration phenomenon is enhanced by the aggregation of powder particles, resulting in the concentration of stress inside the material, and the material is prone to deformation, resulting in a downward trend in the mechanical properties curve of FWPC [4,13]. The application requirements of wood plastic decorative board can be satisfied by the static bending strength (≥20 MPa) and elastic modulus (≥1800 MPa) of FWPC-20 (GB/T 24137-2009, China).

### 3.2. Dynamic Thermomechanical Properties

The curves of FWPC's storage modulus (E′) and loss modulus (E″) are shown in Figure 2, and the characteristic values of loss modulus are shown in Table 2. It can be seen from Figure 2a that as the temperature increases, the FWPC gradually softens, its hardness reduces, and the height of the E′ curve reduces. With the increase in APP/ZB flame-retardant content, the E′ curve reduces. The reason may be that a large number of powder particles (Nano-CB, APP and ZB) are added to the FWPC system, resulting in agglomeration inside the material. The interfacial adhesion between the components in FWPC is reduced, and the hindrance to the thermal motion of the molecular chain is weakened. The rigidity of the material is reduced, resulting in an obvious decrease in E′.

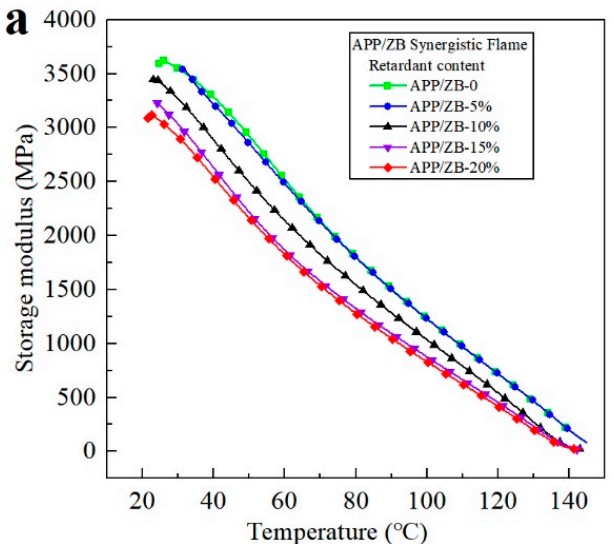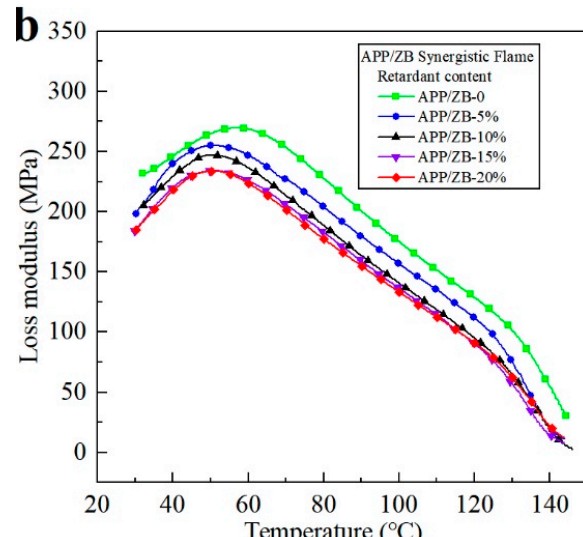

**Figure 2.** Effect of different APP/ZB content on the DMA of FWPC. Effect of different APP/ZB content on storage modulus (**a**) and loss modulus (**b**) under different temperature conditions.

**Table 2.** Effect of APP/ZB content on the characteristic value of FWPC loss modulus.

| Samples | Amount of APP/ZB Added (wt%) | Maximum Internal Consumption (MPa) | Internal Friction Peak Temperature (°C) |
|---|---|---|---|
| FWPC-0 | 0 | 269.01 | 59.30 |
| FWPC-0 | 5 | 255.06 | 50.42 |
| FWPC-0 | 10 | 253.12 | 50.29 |
| FWPC-0 | 15 | 234.04 | 49.75 |
| FWPC-0 | 20 | 233.63 | 49.46 |

As can be seen from Figure 2b and Table 2, with the increase in temperature, E″ first increases and then decreases. The E″ curve shows an internal friction peak caused by α relaxation of regenerated PE-HD between 50 and 60 °C [14]. With the increase in the amount of APP/ZB flame retardant, E″ is reduced, and the temperature of the internal friction peak moves to a lower temperature. When the amount of APP/ZB synergistic flame retardant is from 0 to 20 wt%, the maximum internal friction of FWPC is reduced from 269.01 MPa to 233.63 MPa. This is because a large amount of flame-retardant APP/ZB is added to the composite system, resulting in a weakened compatibility between PE-HD, Salix powder, antistatic agent and flame retardant. The interfacial bonding ability between the components in the wood plastic composite is weakened, and the hindrance to the thermal motion of the molecular chain segment is reduced, resulting in a decrease in heat loss and E″ [15].

### 3.3. Antistatic Properties

The resistivity change trend of FWPC is shown in Figure 3. It can be seen that the surface resistivity of FWPC does not change logarithmically with the increase in APP/ZB synergistic flame retardant. However, the volume resistivity increases slightly. When the content of APP/ZB is 20 wt%, the logarithm of surface resistivity and volume resistivity of FWPC is eight, and the material reaches the antistatic level. The experimental results show that the addition of APP/ZB does not affect the antistatic performance of FWPC, and the antistatic effect is better [16,17].

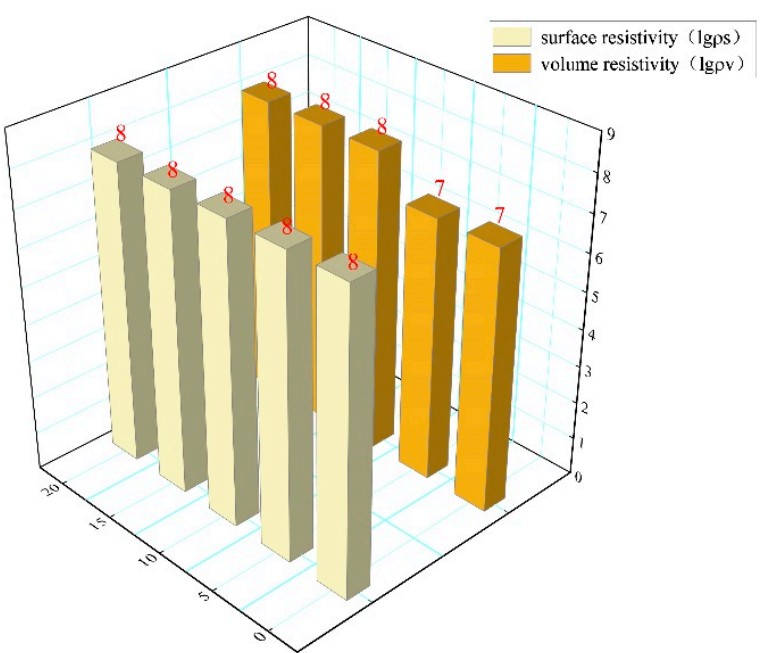

**Figure 3.** Effect of different APP/ZB content on antistatic properties of FWPC.

### 3.4. Thermogravimetric Analysis

With the addition of APP/ZB synergistic flame retardant (from 0 to 20 wt%), the thermal weight loss curve of FWPC is shown in Figure 4, and the characteristic value of the thermal weight loss curve is shown in Table 3. It can be seen from Figure 4 that the TG curve of FWPC can be divided into four stages of weight loss. The first stage (from 0 to 300 °C) is the slow pyrolysis and volatilization process of water and organic additives in FWPC. There is no obvious quality loss in FWPC. The TG curve of the second stage (300~380 °C) shows the obvious weight loss range. This is due to the loss of hemicellulose, cellulose and lignin in wood powder components. At this stage (380~500 °C), PE-HD is heated and cracked. It is broken down into small olefins, alkanes, and other products. At 500~800 °C, the main body of weightlessness is wood flour and PE-HD pyrolysis residue [18].

**Table 3.** Characteristic values of TG curve.

| Samples | Amount of APP/ZB Added (wt%) | The Initial Temperature of Weight Loss in the Second Stage (°C) | Quality Loss Rate (%) | | |
|---|---|---|---|---|---|
| | | | Second Stage (%) | Third Stage (%) | Total Weight Loss Rate (%) |
| FWPC-0 | 0 | 310.01 | 23.38 | 55.58 | 83.71 |
| FWPC-0 | 5 | 288.25 | 21.04 | 52.83 | 75.03 |
| FWPC-0 | 10 | 287.80 | 18.40 | 49.42 | 68.37 |
| FWPC-0 | 15 | 287.45 | 18.27 | 46.87 | 67.51 |
| FWPC-0 | 20 | 286.66 | 17.87 | 42.75 | 61.42 |

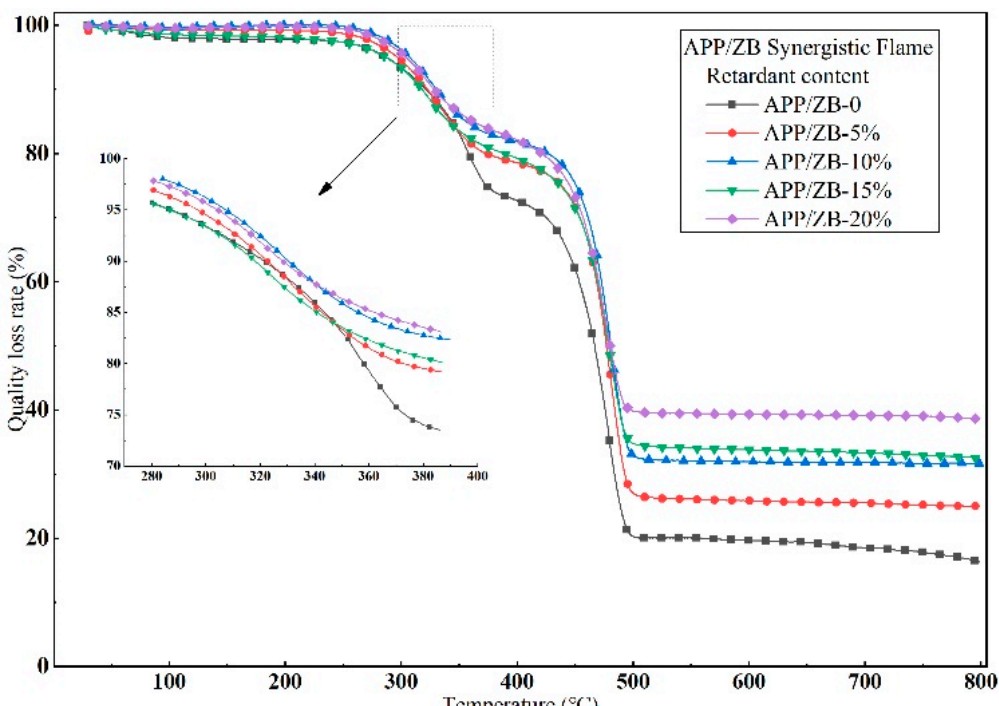

**Figure 4.** Effect of different APP/ZB content on FWPC thermal weight loss.

It can be seen from Table 3 that when the addition from 0 to 20 wt% flame-retardant APP/ZB was conducted to FPWC, the initial temperature of the second stage of weight loss and the weight loss rate of the second and third stages of FWPC showed a downward trend. With the increase in APP/ZB content, the pyrolysis residue quality of FWPC showed an upward trend, and the carbon residue rate increased from 16.29% to 38.58%. The reasons are that the phosphoric acid and free radicals such as HPO and PO are released during the thermal decomposition of APP, which can promote the rapid carbonization of the material. The surface of the composite was covered by ZB heated to form a glassy expansion coating ($B_2O_3$). It can promote the rapid dehydration and carbonization of wood flour, thus reducing the initial temperature and weight loss rate of FWPC [19,20]. When the heat transfer in the composite system is limited by the formed carbon layer, the thermal degradation rate reduces, and the second-order weight loss rate of FWPC reduces [21]. The mass of the final residue was increased, and the thermal stability of FWPC-20 was improved.

### 3.5. Flame Retardant Properties Analysis

The Limit Oxygen Index (LOI) of FWPC is shown in Figure 5. It can be seen that with the increase in the APP/ZB synergistic flame retardant, the LOI value of FWPC increased. The 20 wt% APP/ZB synergistic flame retardant was added to the composite system, and the LOI value of FWPC-20 reached 27.3%. Compared with FWPC-0, the LOI value was increased by 5.30%. The results showed that the flame retardancy of FWPC was significantly improved and reached the level of flame-retardant materials [22].

The main reason for this process was that, when FWPC burns, polyphosphoric acid and poly metaphosphoric acid produced by the thermal decomposition of APP in the system can promote the dehydration of substances into charcoal. The contact between oxygen and material can be diminished by the generated carbon layer. The heat transfer inside the material is limited, and the combustion performance of the material is weakened [23]. At the same time, inert gases such as $H_2O$ and $NH_3$ are released, and the surrounding combustible gases are diluted. Therefore, with the increasing content of APP/ZB in the composite system, the flame retardancy of the material is improved [24,25].

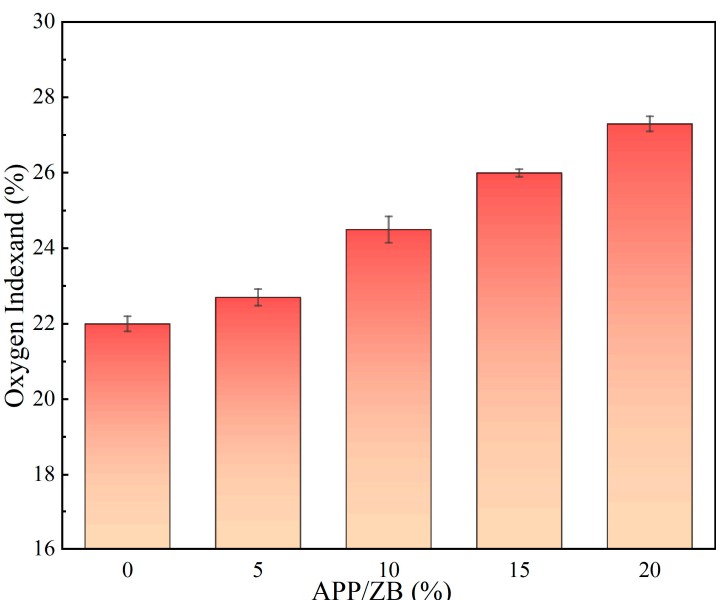

**Figure 5.** Effect of different APP/ZB content on the oxygen index of FWPC.

### 3.6. Cone Calorimeter Test

The relationship of the ignition time (IT), the maximum heat release rate (pk-HRR), the average heat release rate (average HRR), and the total heat release (THR) are shown in Table 4, and the heat release curves are shown in Figure 6. It can be seen from Table 4 and Figure 6 that there were two heat peaks in the heat release curve of FWPC. The first heat peak was a short burning process after the sample was ignited. The second peak was the process in which the flame burned more intensely. With the increase in the content of APP/ZB synergistic flame retardant, the ignition time of FWPC was prolonged. The heat release rate decreased, and the second heat peak appeared a little later. At the same time, the trend flattened and the peak value decreased, which was consistent with the change trend in FWPC total heat release. When the 20 wt% APP/ZB synergistic flame retardant was added to FWPC, the pk-HRR, average HRR and THR of FWPC-20 were 226.75 kw/m$^2$, 110.53 kw/m$^2$ and 66.96 MJ/m$^2$, respectively. Compared with FWPC-0, the above properties were reduced by 60.8%, 51.5% and 30.1%, respectively, and the ignition time was extended from 8 s to 22 s.

**Table 4.** Important parameters of cone calorimetry test for flame-retardant FWPC.

| Samples | Amount of APP/ZB Added (wt%) | IT (s) | pk-HRR (kw/m$^2$) | Average HRR (kw/m$^2$) | THR (600 min) (MJ/m$^2$) |
|---|---|---|---|---|---|
| FWPC-0 | 0 | 8 | 579.69 | 228.15 | 95.78 |
| FWPC-0 | 5 | 12 | 303.28 | 140.30 | 90.09 |
| FWPC-0 | 10 | 21 | 272.60 | 128.35 | 78.66 |
| FWPC-0 | 15 | 22 | 269.74 | 112.65 | 74.93 |
| FWPC-0 | 20 | 22 | 226.75 | 110.53 | 66.96 |

The results show that with the continuous addition of APP/ZB flame retardant, the heating and thermal decomposition rates of FWPC during combustion can be effectively reduced. It may depend on the decomposition of APP in the combustion process to produce polyphosphoric acid and poly metaphosphoric acid, which can promote the dehydration of wood flour to form a carbon layer to a certain extent and produce smoke, thus playing the dual role of solid flame retardant and gas flame retardant [26,27]. When ZB is heated, it decomposes into a glassy oxide film (B$_2$O$_3$) [28]. The transfer of oxygen and heat is isolated by the carbon layer covering the surface of the material and B$_2$O$_3$. Therefore, the heat release rate and total heat release of FWPC-20 were significantly reduced [29].

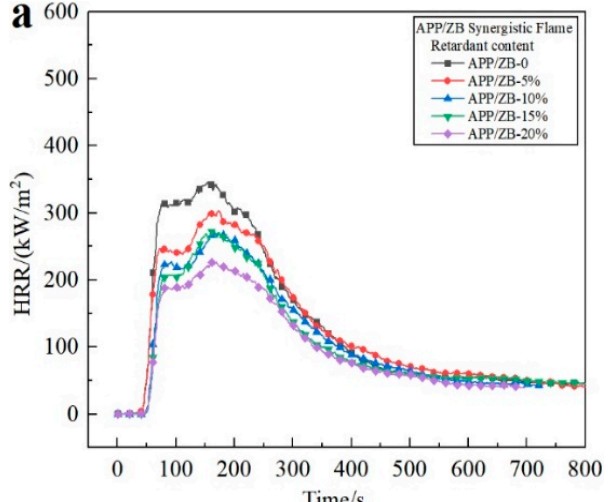
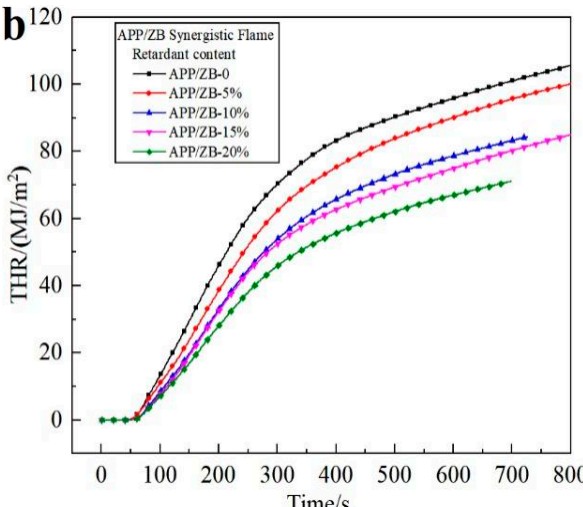

**Figure 6.** Effect of different APP/ZB content on the heat release rate (**a**) and total heat release (**b**) of FWPC.

### 3.7. Micromorphology Analysis

Through the analysis of mechanical properties and flame-retardant properties, it was determined that the optimum addition of APP/ZB synergistic flame retardant was 20%. The tensile section of FWPC with 0 and 20 wt% of APP/ZB synergistic flame retardant was compared and observed. The microstructure was shown in Figure 7a,b. When the content of APP/ZB was 0 and 20 wt%, respectively, the micro-morphology of the carbon layer after FWPC combustion was shown in Figure 7c,d. As can be seen from Figure 7a, a large number of nano-CB particles are attached to the surface of FWPC, and the agglomeration does not occur. It can be seen from Figure 7b that after adding the APP/ZB synergistic flame retardant, not only is a large amount of carbon black distributed on FWPC surface, but also APP/ZB adheres to the surface. A large number of powders were agglomerated in the composite system, which significantly reduced the mechanical properties [30]. It can be seen from Figure 7d that uneven carbon with layered structure appears on the surface of the material. The carbon layer in Figure 7c was thin and distributed in flakes.

The reason was that the flame-retardant APP was decomposed and dehydrated after heating to produce metal phosphate and polyphosphate. The FWPC surface was attached by acidic substances, and the viscosity of FWPC surface increased, forming a large number of carbon layer. The substances produced by ZB decomposition adhere to the material surface, forming a large number of uneven and compact protective layers. It can not only isolate heat and oxygen, but also prevent smoke from overflowing. Therefore, the flame-retardant and smoke-suppression properties of the composites were significantly improved by APP/ZB synergistic flame retardant [31,32].

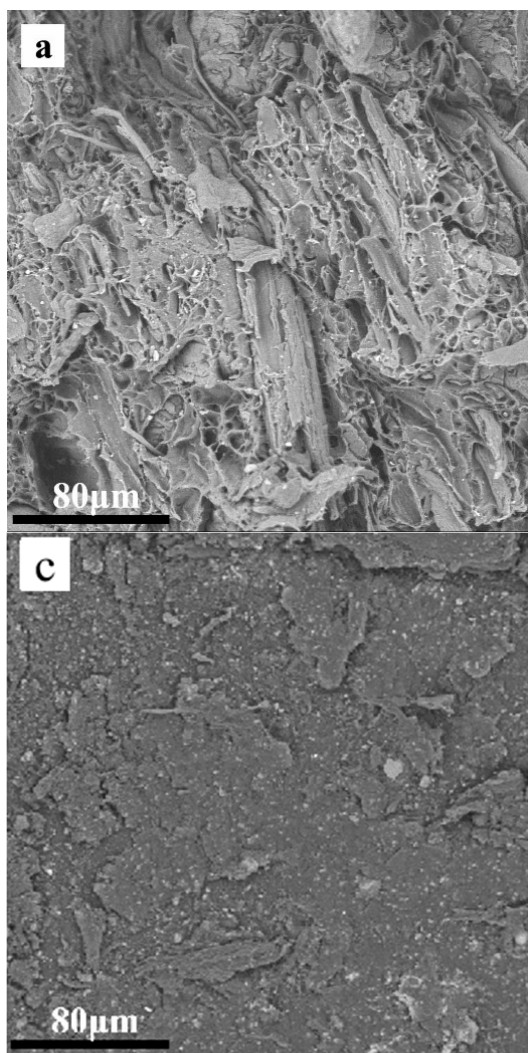
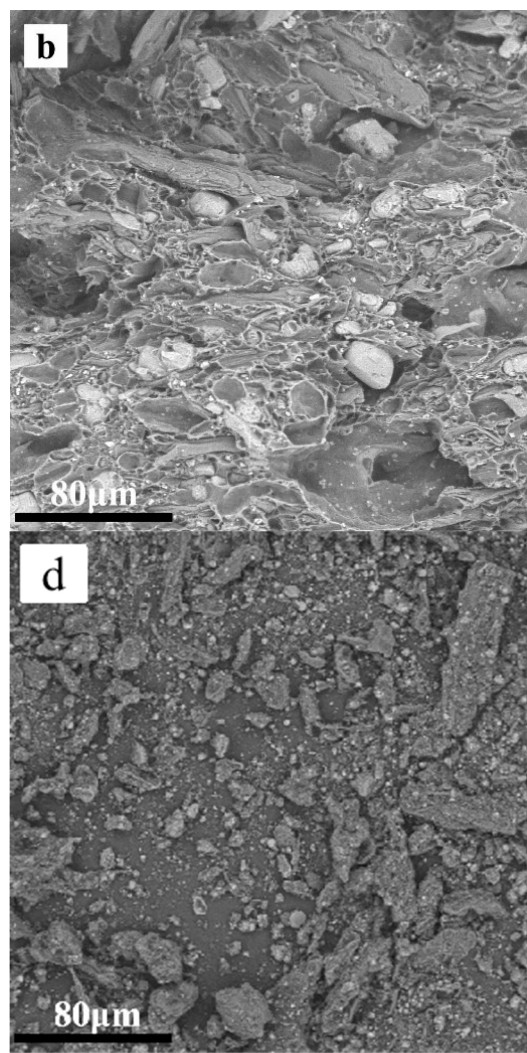

**Figure 7.** (**a**,**b**) are the material interface diagrams of WPC with different APP/ZB additions before combustion (**a**) 0; (**b**) 20 wt%. (**c**,**d**) are the carbon interface diagrams of WPC with different APP/ZB additions after combustion (**c**) 0; (**d**) 20 wt%.

## 4. Conclusions

In this work, a foamed wood–plastic composite with flame retardancy, antistatic properties and high thermal stability was prepared by compression molding. When the flame-retardant APP/ZB is not added, FWPC-0 has the best static bending strength and elastic modulus. When the mass fraction of APP/ZB is 20 wt%, the static bending strength and elastic modulus of FWPC-20 are 30.11 MPa and 2636 MPa, respectively, which can meet the normal use of wood–plastic decorative boards. Due to the poor dispersion of APP/ZB in the composite system, the interfacial bonding force between the components of the material is weakened, resulting in a decrease in its mechanical strength, stiffness and damping properties. FWPC-20 exhibits better thermal stability and flame retardancy (oxygen index value is 27.3%) and has better heat release inhibition ability. At the same time, the addition of nano-carbon black also kept the logarithm of surface resistivity and volume resistivity of FWPC-20 at eight, so that it had good antistatic effect. This study provides a low-cost and simpler method for designing a foamed wood–plastic composite with flame-retardant, antistatic and high mechanical properties. This foamed wood plastic composite material will have broad application prospects in packaging, transportation and other fields.

**Author Contributions:** Z.L.: Investigation, Analysis and collation of data, Writing—initial draft. J.L.: Investigation, Analysis and collation of data, Writing—review and editing. Y.Z.: Formal analysis, revision. X.Z.: Conceptualization, revision. Q.L.: Conceptualization, revision. All authors have read and agreed to the published version of the manuscript.

**Funding:** This work was supported by the Inner Mongolia Natural Science Foundation Project (2022MS03043), Inner Mongolia Autonomous Region Science and Technology Project (2021GG0075), National Natural Science Foundation of China (31660177).

**Institutional Review Board Statement:** Not applicable.

**Informed Consent Statement:** Not applicable.

**Data Availability Statement:** The data presented in this study are available on request from the corresponding author.

**Conflicts of Interest:** The authors declare no conflict of interest.

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
