# Peer review of "Preparation and Properties of Flame Retardant and Antistatic Foamed Wood–Plastic Composite with APP/ZB System"

_coatings, doi:10.3390/coatings13040789_

Round 1

Reviewer 1 Report

The topic of this work is really interesting and provides some new information to the field. Nevertheless, there are some points to be improved or corrected in the text of the manuscript. You should leave spaces between the value and the unit. In the abstract, it is not appropriate to present almost all the mean values of the results, just referring the values, without discussing the interpretation and potential utilization of them in future. The sentence in lines 46-47 needs improvement from the point of grammar. The state-of-the art is quite well provided and analyzed. In line 58, AA that is referred has not been previously explained in the text, please provide the explanation. There is not exaplanation and the providance of any infromation related to the raw wood material. Did it correpond to one or more trunks? what was the age of them? which was the moisture content at the time of use in the composites? was it sapwood, heartwood or both. You did not refer to the reason why you have chosen the specific species to be studied in such an application. Provide information concerning the "universal prototype machine". Please, provide all the necessary details on the equipment used (country, manufacturer model etc.). There is not any description of statistical analysis, is it possible? Provide the standard deviation values of the results in the graphs. In 307 line 20% is not clear on the way that is meant.

Author Response

Major revision taking into account the response point by point for the following:

Reviewer #1:

  1. The topic of this work is really interesting and provides some new information to the field. Nevertheless, there are some points to be improved or corrected in the text of the manuscript. You should leave spaces between the value and the unit. In the abstract, it is not appropriate to present almost all the mean values of the results, just referring the values, without discussing the interpretation and potential utilization of them in future.

Thanks a lot for your advice. Based on your advice, the part had been revised in revised manuscript.

All spaces between values and units have been reserved in the revised manuscript.

Based on your advice, The abstract has been modified as follows:

With the aggravation of fire and smoke pollution, it is urgent to develop green, lower cost and high-performance Foamed Wood-Plastic Composite (FWPC) to meet the standards of antistatic, flame retardant and fire prevention in practical application. Therefore, the flame retardant and antistatic FWPC were prepared by compression molding in this study. High-density polyethylene (PE-HD) and Salix wood-flour were used as main raw materials, and azodicarbonamide (AC) was used as foaming agent, Nano-carbon black (Nano-CB) as antistatic filler, ammonium polyphosphate (APP) and zinc borate (ZB) as flame retardants. The static bending strength and elastic modulus of FWPC-20 reached 30.01 MPa and 2636 MPa, respectively, which can meet the commercial application of wood-plastic decorative board. The logarithm of surface resistivity and volume resistivity of FWPC-20 is kept at 8, indicating that it has antistatic effect. The residual carbon rate of FWPC-20 increased to 38.58 % at 800°C, indicating that FWPC had high thermal stability. The minimum heat release rate of FWPC-20 was 226.75 kw/m2, the average heat release rate was 110.53 kw/m2, the total heat release was 66.96 MJ/m2, and the limit oxygen index was 27.3 %, which indicated that FWPC-20 had flame retardant and smoke suppression effects. This study provides a low-cost and simple method for the design of flame retardant, antistatic and high-performance FWPC, and has broad application prospects in the fields of packaging and construction.

  1. The sentence in lines 46-47 needs improvement from the point of grammar. The state-of-the art is quite well provided and analyzed.

Thanks a lot for your advice. Based on your advice, the part had been revised in revised manuscript.

In recent years, related wood-plastic composites have been extensively studied by researchers. Zhang et al[1] prepared a biochar/wood-plastic composite by extrusion molding, and tested its mechanical properties and flame retardancy. The results showed that with the increase of biochar content, the mechanical properties of wood-plastic composites increased first and then decreased. The addition of Mg(OH)2 and Al(OH)3 can significantly improve the flame retardant properties of the compo-sites. When the addition of Mg(OH)2 is 40wt %, the flame retardant effect of the com-posites is the best.

Reference

[1] Q F Zhang, H Z Cai, K Y Yang, et al. Effect of biochar on mechanical and flame retardant properties of wood–Plastic composites[J]. Results in Physics, 2017, 7: 2391-2395.

  1. In line 58, AA that is referred has not been previously explained in the text, please provide the explanation.

Thanks a lot for your advice. AA is the abbreviation of antistatic agent.

  1. There is not exaplanation and the providance of any infromation related to the raw wood material. Did it correpond to one or more trunks?

Thanks a lot for your advice. Based on your advice, the part had been revised in revised manuscript.

Salix wood-flour (the particle size is 80~100 mesh) was supplied by Man Lai Township Ordos City Inner Mongolia, China (It has both sapwood and heartwood. The age of Salix is 5-6 years. The moisture content is 2 wt%).

  1. what was the age of them?

Thanks a lot for your advice. The age of Salix is 5-6 years.

  1. which was the moisture content at the time of use in the composites?

Thanks a lot for your advice. The moisture content is 2 wt%.

  1. was it sapwood, heartwood or both.

Thanks a lot for your advice. It has both sapwood and heartwood.

  1. You did not refer to the reason why you have chosen the specific species to be studied in such an application.

Thanks a lot for your advice. Based on your advice, the reasons for choosing Salix for research are as follows.

Salix has many advantages such as renewable energy, wide source and low cost. Wood plastic composites were prepared by mixing salix and resin. While strengthening and modifying the resin, it has opened up a way for high-value utilization of Salix resources.

  1. Provide information concerning the "universal prototype machine". Please, provide all the necessary details on the equipment used (country, manufacturer model etc.). There is not any description of statistical analysis, is it possible?

Thanks a lot for your advice. Based on your advice, the part had been revised in revised manuscript.

The static bending strength and elastic modulus were tested by Electronic universal testing machine (Model, WDW-20A, Jinan Tianchen Machinery Manufacturing Co., Ltd., China). The tensile strength was tested by a Mechanical testing machine (Model, AG-IC, Shimadzu Instruments (Suzhou) Co., Ltd. China). The impact strength was tested by a Pendulum Impact Tester (Model, ZBC7151-B, Meters Industrial Systems Co., Ltd. China). Dynamic mechanical properties of the composites were measured by using a dynamic mechanical analyser (Model Q800, TA company, USA) under Multi-Frequency-Strain mode. Oxygen index of the samples were measured by Oxygen Index Meter (Model, JF-3, Nanjing Jiangning District Analysis Instrument Factory, China). The electrical conductivity of FWPC was tested by a resistivity tester (Model, BEST-212, Beijing Beiguang Precision Instrument Equipment Co., Ltd. China). Thermal degradation of the composites was studied by a thermogravimetric analyzer, (STA409PC, NETZSCH Germany). The reaction-to-fire of FWPC was tested by (FTT) cone calorimeter (Model: MOTS, Motis Company, China). Microstructure images of samples were obtained by scanning electron microscopy (Phenom Pro, Funa Scientific Instruments (Shanghai) Co., Ltd. Holland).

  1. Provide the standard deviation values of the results in the graphs.

Thanks a lot for your advice. The graphics that can provide standard deviation have been marked in the revised draft.

Figure 1. Effect of different APP/ZB content on static bending strength and elastic modulus (a). Effect of different APP/ZB content on tensile strength and impact strength (b)

Figure 2. Effect of different APP/ZB content on the oxygen index of FWPC.

  1. In 307 line 20% is not clear on the way that is meant.

Thanks a lot for your advice. Based on your advice, the part had been revised in revised manuscript.

In this work, a foamed wood-plastic composite with flame retardancy, antistatic properties and high thermal stability was prepared by compression molding. When the flame retardant APP/ZB is not added, FWPC-0 has the best static bending strength and elastic modulus. When the mass fraction of APP/ZB is 20 wt%, the static bending strength and elastic modulus of FWPC-20 are 30.11 MPa and 2636 MPa, respectively, which can meet the normal use of wood plastic decorative boards. Due to the poor dispersion of APP/ZB in the composite system, the interfacial bonding force between the components of the material is weakened, resulting in a decrease in its mechanical strength, stiffness and damping properties. However, FWPC-20 exhibits better thermal stability and flame retardancy (oxygen index value is 27.3 %), and has better heat release inhibition ability. At the same time, the addition of nano-carbon black also kept the logarithm of surface resistivity and volume resistivity of FWPC-20 at 8, so that it had good antistatic effect. This study provides a low-cost and simpler method for designing a foamed wood-plastic composite with flame retardant, antistatic and high mechanical properties. This foamed wood plastic composite material will have broad application prospects in packaging, transportation and other fields.

Reviewer 2 Report

It is not clear how the work presented in this manuscript has any relevance to the aims of the Coatings journal. It should be submitted to another journal that focuses on bulk modifications. The English of the manuscript is very poor. The first sentence of the abstract is incomplete and many similar errors exist throughout the manuscript. It is not clear what the rationale behind this work is, the motivation and state of art are missing or not clearly mentioned. How can the authors claim APP/ZB system is synergistic? Seems, the authors have collected different ideas from the literature and prepared a technical report of the experiments they have performed.

Author Response

Major revision taking into account the response point by point for the following:

Reviewer #2:

  1. It is not clear how the work presented in this manuscript has any relevance to the aims of the Coatings journal. It should be submitted to another journal that focuses on bulk modifications. The English of the manuscript is very poor.

Thanks a lot for your advice. 

Coating journals are committed to publishing functional, protective and decorative flame retardant coating materials. This study is devoted to the study of fire retardant coating materials, which is consistent with the theme of the journal.

English has been polished in revised manuscript.

  1. The first sentence of the abstract is incomplete and many similar errors exist throughout the manuscript.

Thanks a lot for your advice. Based on your advice, the part had been revised in revised manuscript. 

With the aggravation of fire and smoke pollution, it is urgent to develop green, lower cost and high-performance Foamed Wood-Plastic Composite (FWPC) to meet the standards of antistatic, flame retardant and fire prevention in practical application. Therefore, the flame retardant and antistatic FWPC were prepared by compression molding in this study. High-density polyethylene (PE-HD) and Salix wood-flour were used as main raw materials, and azodicarbonamide (AC) was used as foaming agent, Nano-carbon black (Nano-CB) as antistatic filler, ammonium polyphosphate (APP) and zinc borate (ZB) as flame retardants. The static bending strength and elastic modulus of FWPC-20 reached 30.01 MPa and 2636 MPa, respectively, which can meet the commercial application of wood-plastic decorative board. The logarithm of surface resistivity and volume resistivity of FWPC-20 is kept at 8, indicating that it has antistatic effect. The residual carbon rate of FWPC-20 increased to 38.58 % at 800 °C, indicating that FWPC had high thermal stability. The minimum heat release rate of FWPC-20 was 226.75 kw/m2, the average heat release rate was 110.53 kw/m2, the total heat release was 66.96 MJ/m2, and the limit oxygen index was 27.3 %, which indicated that FWPC-20 had flame retardant and smoke suppression effects. This study provides a low-cost and simple method for the design of flame retardant, antistatic and high-performance FWPC, and has broad application prospects in the fields of packaging and construction.

  1. It is not clear what the rationale behind this work is, the motivation and state of art are missing or not clearly mentioned.

Thanks a lot for your advice. Based on your advice, the part had been revised in revised manuscript.

With the aggravation of fire and smoke pollution, it is urgent to develop green, lower cost and high-performance Foamed Wood-Plastic Composite (FWPC) to meet the standards of antistatic, flame retardant and fire prevention in practical application. Therefore, the flame retardant and antistatic FWPC were prepared by compression molding in this study.

  1. How can the authors claim APP/ZB system is synergistic?

Thanks a lot for your advice. Based on your advice, the part had been revised in revised manuscript.

APP has the best flame retardant effect, but it cannot inhibit the release of harmful gases. During the FWPC combustion process, carbon monoxide, carbon dioxide and other gases can be reduced by ZB. If the two are combined, FWPC is flame retardant and inhibits harmful gas emissions [1].

Reference

[1] RAMAZAN KURT, FATİH MENGELOĞLU. Utilization of boron compounds as synergists with ammonium polyphosphate for flame retardant wood-polymer composites[J]. TURKISH JOURNAL OF AGRICULTURE AND FORESTRY, 2011, 35(2):155-163.

  1. Seems, the authors have collected different ideas from the literature and prepared a technical report of the experiments they have performed.

Thanks a lot for your advice. Based on your advice, the part had been revised in revised manuscript.

Reviewer 3 Report

I attached my comments.

Author Response

Major revision taking into account the response point by point for the following:

Reviewer #3:

  1. 1.Abstract, r.11: “In order to make wood-plastic materials to meet antistatic, flame retardant, and fire protection standards in practical applications.” – There is no action verb, it has no sense.

Thanks a lot for your advice. The abstract part has been modified, and the results are as follows.

With the aggravation of fire and smoke pollution, it is urgent to develop green, lower cost and high-performance Foamed Wood-Plastic Composite (FWPC) to meet the standards of antistatic, flame retardant and fire prevention in practical application. Therefore, the flame retardant and antistatic FWPC were prepared by compression molding in this study. High-density polyethylene (PE-HD) and Salix wood-flour were used as main raw materials, and azodicarbonamide (AC) was used as foaming agent, Nano-carbon black (Nano-CB) as antistatic filler, ammonium polyphosphate (APP) and zinc borate (ZB) as flame retardants. The static bending strength and elastic modulus of FWPC-20 reached 30.01 MPa and 2636 MPa, respectively, which can meet the commercial application of wood-plastic decorative board. The logarithm of surface resistivity and volume resistivity of FWPC-20 is kept at 8, indicating that it has antistatic effect. The residual carbon rate of FWPC-20 increased to 38.58 % at 800 °C, indicating that FWPC had high thermal stability. The minimum heat release rate of FWPC-20 was 226.75 kw/m2, the average heat release rate was 110.53 kw/m2, the total heat release was 66.96 MJ/m2, and the limit oxygen index was 27.3 %, which indicated that FWPC-20 had flame retardant and smoke suppression effects. This study provides a low-cost and simple method for the design of flame retardant, antistatic and high-performance FWPC, and has broad application prospects in the fields of packaging and construction.

  1. Abstract, r.16: The properties are studied by different methods, like calorimetry, microscopy, using different equipments, like calorimeter or microscope. Please reformulate accordingly.

Thanks a lot for your advice. The abstract part has been modified, and the results are as follows.

With the aggravation of fire and smoke pollution, it is urgent to develop green, lower cost and high-performance Foamed Wood-Plastic Composite (FWPC)  to meet the standards of antistatic, flame retardant and fire prevention in practical application. Therefore, the flame retardant and antistatic FWPC were prepared by compression molding in this study. High-density polyethylene (PE-HD) and Salix wood-flour were used as main raw materials, and azodicarbonamide (AC) was used as foaming agent, Nano-carbon black (Nano-CB) as antistatic filler, ammonium polyphosphate (APP) and zinc borate (ZB) as flame retardants. The static bending strength and elastic modulus of FWPC-20 reached 30.01 MPa and 2636 MPa, respectively, which can meet the commercial application of wood-plastic decorative board. The logarithm of surface resistivity and volume resistivity of FWPC-20 is kept at 8, indicating that it has antistatic effect. The residual carbon rate of FWPC-20 increased to 38.58 % at 800 °C, indicating that FWPC had high thermal stability. The minimum heat release rate of FWPC-20 was 226.75 kw/m2, the average heat release rate was 110.53 kw/m2, the total heat release was 66.96 MJ/m2, and the limit oxygen index was 27.3 %, which indicated that FWPC-20 had flame retardant and smoke suppression effects. This study provides a low-cost and simple method for the design of flame retardant, antistatic and high-performance FWPC, and has broad application prospects in the fields of packaging and construction.

  1. Abstract, r.22: “compared with no addition” ??????

Thanks a lot for your advice. The abstract part has been modified, and the results are as follows.

With the aggravation of fire and smoke pollution, it is urgent to develop green, lower cost and high-performance Foamed Wood-Plastic Composite (FWPC)  to meet the standards of antistatic, flame retardant and fire prevention in practical application. Therefore, the flame retardant and antistatic FWPC were prepared by compression molding in this study. High-density polyethylene (PE-HD) and Salix wood-flour were used as main raw materials, and azodicarbonamide (AC) was used as foaming agent, Nano-carbon black (Nano-CB) as antistatic filler, ammonium polyphosphate (APP) and zinc borate (ZB) as flame retardants. The static bending strength and elastic modulus of FWPC-20 reached 30.01 MPa and 2636 MPa, respectively, which can meet the commercial application of wood-plastic decorative board. The logarithm of surface resistivity and volume resistivity of FWPC-20 is kept at 8, indicating that it has antistatic effect. The residual carbon rate of FWPC-20 increased to 38.58 % at 800 °C, indicating that FWPC had high thermal stability. The minimum heat release rate of FWPC-20 was 226.75 kw/m2, the average heat release rate was 110.53 kw/m2, the total heat release was 66.96 MJ/m2, and the limit oxygen index was 27.3 %, which indicated that FWPC-20 had flame retardant and smoke suppression effects. This study provides a low-cost and simple method for the design of flame retardant, antistatic and high-performance FWPC, and has broad application prospects in the fields of packaging and construction.

  1. r.47: “the best”

Thanks a lot for your advice. Based on your advice, the part had been revised in revised manuscript.

In recent years, related wood-plastic composites have been extensively studied by researchers. Zhang et al prepared a biochar/wood-plastic composite by extrusion molding, and tested its mechanical properties and flame retardancy. The results showed that with the increase of biochar content, the mechanical properties of wood-plastic composites increased first and then decreased. The addition of Mg(OH)2 and Al(OH)3 can significantly improve the flame retardant properties of the composites. When the addition of Mg(OH)2 is 40wt %, the flame retardant effect of the composites is the best.

  1. r.49: “was investigated

Thanks a lot for your advice. Based on your advice, the part had been revised in revised manuscript.

In recent years, related wood-plastic composites have been extensively studied by researchers. Zhang et al. prepared a biochar/wood-plastic composite by extrusion molding, and tested its mechanical properties and flame retardancy. The results showed that with the increase of biochar content, the mechanical properties of wood-plastic composites increased first and then decreased. The addition of Mg(OH)2 and Al(OH)3 can significantly improve the flame retardant properties of the composites. When the addition of Mg(OH)2 is 40wt %, the flame retardant effect of the composites is the best.

  1. r.52: “were obviously enhanced.”

Thanks a lot for your advice. In response to this problem, I have corrected it and the results are as follows.

In recent years, related wood-plastic composites have been extensively studied by researchers. Zhang et al. prepared a biochar/wood-plastic composite by extrusion molding, and tested its mechanical properties and flame retardancy. The results showed that with the increase of biochar content, the mechanical properties of wood-plastic composites increased first and then decreased. The addition of Mg(OH)2 and Al(OH)3 can significantly improve the flame retardant properties of the composites. When the addition of Mg(OH)2 is 40wt %, the flame retardant effect of the composites is the best.

  1. r.52: “PE-HD/Salix composite was an electrical insulator because of its high electrical resistivity. After friction, a large amount of static electricity will be generated on the surface of wood-plastic composites. It will cause electrostatic leakage and electromagnetic interference” - What is the relevance of this phrase here?

Thanks a lot for your advice. Here I mainly want to emphasize the harm of electrostatic leakage.

  1. r.58: AA ????

Thanks a lot for your advice. AA is Antistatic agents

  1. r.63: What is CONE test???

Thanks a lot for your advice. CONE test is Cone calorimeter test [1].

Reference

[1] Qingfeng Xu et al. Combustion and charring properties of five common constructional wood species from cone calorimeter tests[J]. Construction and Building Materials, 2015, 96 : 416-427. DOI: 10.1016/j.conbuildmat.2015.08.062

  1. r.67: “High-efficient flame retardant-ammonium polyphosphate with the molecular formula of (NH4)n+2PnO3n+1.” - There is no action verb, it has no sense.

Thanks a lot for your advice. I have modified this sentence and the results are as follows.

High-efficient flame retardant ammonium polyphosphate with the molecular formula is called (NH4)n+2PnO3n+1.

  1. r.79: “Based on the above reasons, this paper focuses on the research that using PE-HD, Salix wood-flour as the main base material, nano-carbon black (Nano-CB) as conductive agent, Ammonium polyphosphate (APP) and zinc borate (ZB) as synergistic flame retardants to create flame retardant and antistatic foamed wood-plastic composites by melt blending.” - ...on the research that using...to create... There is no meaning, it needs to be rewritten correctly.

Thanks a lot for your advice. This problem has been modified and the results are as follows.

In this study, azodicarbonamide (AC) was used as foaming agent, zinc oxide (ZnO) was used as initiator, and nano-calcium carbonate (Nano-CaCO3) was used as nucleating agent. The foaming process effectively reduced the density and weight of Salix/PE-HD composites, and at the same time, the cell could effectively prevent crack propagation and improve the poor toughness of the composites. In addition, nano conductive carbon black (Nano-CB) and synergistic flame retardant APP/ZB were added to FWPC system to prepare flame retardant and antistatic Salix wood-flour/PE-HD foamed composites. The combination of APP and ZB was used as synergistic flame retardant, which made FWPC have good flame retardant and smoke suppression performance on the basis of maintaining mechanical properties, and real-ized the goal of functionalization and high value of wood-plastic composites, which had important practical significance for broadening the application of composites.

  1. r.103: There is no need of 3 paragraphs for describing the preparation of wood-plastic composites. So, paragraphs 2.2, 2.3 and 2.4 can be merged in one paragraph named Preparation of wood-plastic composites. Moreover, the entire paragraph must be rewritten in a more clearly and correct manner. How come the sum of the mass of PE-HD and Salix…was the total mass of FWPC, when obviously there are other important additives in the prepared composites, in amounts that exceed 20%?

Thanks a lot for your advice. The preparation method of wood plastic composites has been rewritten, and the results are as follows.

Firstly, the Salix powder was dried and modified with 12 wt% NaOH solution at 60 °C for 60 min, and then washed with distilled water to neutral. KH550 was hydrolyzed with 95 wt% ethanol, mixed with modified wood flour and Nano-CB respectively, and dried for later use. The temperature of the double roller of the mixer was raised to 160 °C and 170 °C, and PE-HD, related foaming additives, modified wood powder, modified Nano-CB and flame retardant were added in turn for uniform mixing. Finally, the mixture was chopped and put into the mold, which was formed by hot pressing and cold pressing. The hot pressing temperature, pressure and time were 190 °C, 5 MPa and 7 min, respectively. The cold pressing pressure and time were 7 MPa and 10 min.

The sum of the mass of PE-HD and Salix is the total mass of FWPC for the following reasons. Because the addition of ZnO was 20 wt% (mass percentage of foaming agent AC). The addition of AC was 1.5 wt% (Percentage of FWPC total mass).

  1. r.109: “Compound ammonium polyphosphate (APP) and zinc borate (ZB) in a ratio (4:1), the content of APP/ZB synergistic flame retardant was 0, 5%, 10%, 15% and 20%.” – There is no verb here!!!!

Thanks a lot for your advice. This problem has been modified and the results are as follows.

Composite ammonium polyphosphate (APP) and zinc borate (ZB) were added in proportion (4:1), and the amount of APP/ZB synergistic flame retardant was 0, 5 wt%, 10 wt%, 15 wt% and 20 wt%, respectively. The obtained FWPCs were named FWPC-0, FWPC-5, FWPC-10, FWPC-15, FWPC-20, respectively. (The following values of FWPC represent the amount of APP/ZB added).

  1. r.116: “Hydrolysis of KH 550 with 95% ethanol with the mass ratio of 5: 1. Mix with alkali-treated wood flour and Nano-CB, and drying for standby.” Must be reformulated!

Thanks a lot for your advice. This problem has been modified and the results are as follows.

Firstly, the Salix powder was dried and modified with 12 wt% NaOH solution at 60 °C for 60 min, and then washed with distilled water to neutral. KH550 was hydrolyzed with 95 wt% ethanol, mixed with modified wood flour and Nano-CB respectively, and dried for later use. The temperature of the double roller of the mixer was raised to 160 °C and 170 °C, and PE-HD, related foaming additives, modified wood powder, modified Nano-CB and flame retardant were added in turn for uniform mixing. Finally, the mixture was chopped and put into the mold, which was formed by hot pressing and cold pressing. The hot pressing temperature, pressure and time were 190 °C, 5 MPa and 7 min, respectively. The cold pressing pressure and time were 7 MPa and 10min.

  1. r.119: “The above experimental raw materials were prepared…” – the raw materials do not need to be prepared that is why they are raw materials. Maybe they need to be processed or subjected to different treatments in order to obtain some products from them.

Thanks a lot for your advice. The reasons for this problem are as follows. They need to be processed or subjected to different treatments. Wood flour was treated by coupling agent and alkaline solution. Nano-CB was treated with coupling agent. PE-HD, foaming agent and flame retardant were dried.

  1. r.125: Paragraph entitled “Characterization” (of what???) also needs extensive corrections.

Thanks a lot for your advice. Based on your advice, the part had been revised in revised manuscript.

The static bending strength and elastic modulus were tested by Electronic universal testing machine (Model, WDW-20A, Jinan Tianchen Machinery Manufacturing Co., Ltd., China). The tensile strength was tested by a Mechanical testing machine (Model, AG-IC, Shimadzu Instruments (Suzhou) Co., Ltd. China). The impact strength was tested by a Pendulum Impact Tester (Model, ZBC7151-B, Meters Industrial Systems Co., Ltd. China). Dynamic mechanical properties of the composites were measured by using a dynamic mechanical analyser (Model Q800, TA company, USA) under Multi-Frequency-Strain mode. Oxygen index of the samples were measured by Oxygen Index Meter (Model, JF-3, Nanjing Jiangning District Analysis Instrument Factory, China). The electrical conductivity of FWPC was tested by a resistivity tester (Model, BEST-212, Beijing Bei guang Precision Instrument Equipment Co., Ltd. China). Thermal degradation of the composites was studied by a thermogravimetric analyzer, (STA409PC, NETZSCH Germany). The reaction-to-fire of FWPC was tested by (FTT) cone calorimeter (Model: MOTS, Motis Company, China). Microstructure images of samples were obtained by scanning electron microscopy (Phenom Pro, Fu na Scientific Instruments (Shanghai) Co., Ltd. Holland).

  1. r.126: “…elastic modulus of the samples…” as there were more samples.

Thanks a lot for your advice. We selected three samples for three tests, and then calculated the average value.

  1. r.128: “The tensile strength of the samples…was measured…” not the tensile strength test were measured. And so on, there are so many mistakes here, please correct them all.

Thanks a lot for your advice. Based on your advice, the part had been revised in revised manuscript.

The static bending strength and elastic modulus were tested by Electronic universal testing machine (Model, WDW-20A, Jinan Tianchen Machinery Manufacturing Co., Ltd., China). The tensile strength was tested by a Mechanical testing machine (Model, AG-IC, Shimadzu Instruments (Suzhou) Co., Ltd. China). The impact strength was tested by a Pendulum Impact Tester (Model, ZBC7151-B, Meters Industrial Systems Co., Ltd. China). Dynamic mechanical properties of the composites were measured by using a dynamic mechanical analyser (Model Q800, TA company, USA) under Multi-Frequency-Strain mode. Oxygen index of the samples were measured by Oxygen Index Meter (Model, JF-3, Nanjing Jiangning District Analysis Instrument Factory, China). The electrical conductivity of FWPC was tested by a resistivity tester (Model, BEST-212, Beijing Beiguang Precision Instrument Equipment Co., Ltd. China). Thermal degradation of the composites was studied by a thermogravimetric analyzer, (STA409PC, NETZSCH Germany). The reaction-to-fire of FWPC was tested by (FTT) cone calorimeter (Model: MOTS, Motis Company, China). Microstructure images of samples were obtained by scanning electron microscopy (Phenom Pro, Funa Scientific Instruments (Shanghai) Co., Ltd. Holland).

  1. r.135: Lol test ?????

Thanks a lot for your advice. LoI test is the limit oxygen index test [1-2].

Reference

[1] Chen He, et al. Effects of APP/SiO2 polyelectrolyte composites on wood-plastic composite [J].MATEC Web of Conferences, 2019, 275: 01004. DOI:10.1051/matecconf/201927501004

[2] Zvonimir Katančić, et al. Effect of modified nanofillers on fire retarded high-density polyethylene/wood composites[J]. Journal of Composite Materials, 2014, 48(30): 3771-3783. DOI:10.1177/0021998313514085

  1. r.138: remove “Antistatic properties.”

Thanks a lot for your advice. Based on your advice, the part had been revised in revised manuscript.

  1. r.140: TG ???

Thanks a lot for your advice. TG is the abbreviation of thermogravimetric analysis[1-2].

Reference

[1] Wang Wei et al. Thermogravimetric Analysis and Kinetic Modeling of the AAEM-Catalyzed Pyrolysis of Woody Biomass[J]. Molecules, 2022, 27(22): 7662-7662. 

DOI: 10.3390/MOLECULES27227662

  1. r.148: “After spraying…., observe the morphology” – needs reformulation.

Thanks a lot for your advice. This problem has been modified and the results are as follows.

Microstructure images of samples were obtained by scanning electron microscopy (Phenom Pro, Funa Scientific Instruments (Shanghai) Co., Ltd. Holland).

  1. r.159: What does it mean “the retention rates”?

Thanks a lot for your advice. The retention rate refers to the ratio of the reduced mechanical properties of FWPC to the initial properties.

  1. r.160: APP and ZB are they high polar polymers?

Thanks a lot for your advice. Based on your advice, the part had been revised in revised manuscript.

APP has polarity. Zinc borate has no polarity. This is because the polarity of APP/ZB and PE-HD matrix were different, which leads to their compatibility being destroyed[1].

Reference

[1] Xin feng Wu, et al. Flammability of EVA/IFR (APP/PER/ZB system) and EVA/IFR/synergist (CaCO3, NG, and EG) composites[J]. Journal of Applied Polymer Science, 2012, 126(6):1917-1928. DOI:10.1002/app.36884

  1. r.204: The TG curves are not well explained. In my opinion the 3 decomposition stages are: 0-3000C for water and organic additives loss, second stage: 300-380for wood decomposition, and third stage: 380-500for PE depolymerization. I also don’t understand what is “quality loss rate”.

Thanks a lot for your advice. Based on your advice, the part had been revised in revised manuscript.

The first stage (0~300 ℃) is the slow pyrolysis and volatilization process of water and organic additives in FWPC. There is no obvious quality loss in FWPC. The TG curve of the second stage (300~380 ℃) shows obvious weight loss range. This is due to the loss of hemicellulose, cellulose and lignin in wood powder components. At this stage (380~500 ℃), PE-HD is heated and cracked. It is broken down into small olefins, alkanes and other products. At 500 ~ 800 ℃, the main body of weightlessness is wood flour and PE-HD pyrolysis residue.

The mass loss rate refers to the percentage of the sample lost by pyrolysis volatilization.

  1. “The quality of pyrolysis was on rise…”????

Thanks a lot for your advice. Based on your advice, the part had been revised in revised manuscript.

With the increase of APP/ZB content, the pyrolysis residue quality of FWPC showed an upward trend, and the carbon residue rate increased from 16.29 % to 38.58 %.

  1. r.282: I don’t think that mechanical parameters recommend the addition of 20% of APP/ZB as in this case the composites have the lower mechanical performances. So, I suggest reformulating this sentence.

Thanks a lot for your advice. In response to your question, I made the following explanation.

When the amount of APP / ZB is 20 wt%, although the mechanical properties of FWPC are low, the material has better thermal stability, flame retardancy and heat release inhibition.

  1. r.287: “It can be seen from Figure 7 (a) that the synergistic flame retardant without APP/ZB only adheres to a large number of nano-CB particles, and no obvious agglomeration was observed.” What adheres to a large number of CB particles if there is no APP/ZB present?? Another sentence that is difficult to understand.

Thanks a lot for your advice. Based on your advice, the part had been revised in revised manuscript.

As can be seen from Figure. 7 (a), a large number of nano-CB particles are attached to the surface of FWPC, and no agglomeration occurs.

  1. r.293: “carbon layer … was granular distribution and expansion” ???

Thanks a lot for your advice. The carbon layer is a uneven layered structure. I have changed and marked red in the revised manuscript.

  1. r.296: The legend of Figure 7 is not complete, there is no reference to (a)-(d) corresponding images.

Thanks a lot for your advice. Based on your advice, the part had been revised in revised manuscript.

Figure 7. a-b is the material interface diagram of WPC with different APP/ZB addition before combustion (a: 0; b: 20 wt%). c-d is the carbon interface diagram of WPC with different APP/ZB ad-ditions after combustion (c: 0; d: 20 wt%).

  1. r.299: How was the surface of FWPC attached by acidic substances and the surface viscosity increased?

Thanks a lot for your advice. Because APP / ZB has been uniformly mixed into FWPC during the experiment. Therefore, APP / ZB can be observed on the surface of the carbon layer [1].

Reference

[1] Zhao wen Liu, et al. Synthesis of a triazine-based macromolecular hybrid charring agent containing zinc borate and its flame retardancy and thermal properties in polypropylene[J]. International Journal of Polymer Analysis and Characterization, 2020, 25(5): 334-342. DOI:10.1080/1023666X.2020.1786790

Round 2

Reviewer 1 Report

After the thorough check of the revised text, I have seen that the authors have implemented most of the proposed changes in the revised verion of manuscript towards the improvement of their work. Almost all the changes have been implemented and in my opinion, but a final check should be imeplemented towards the correction of some syntactical/grammatical errors in the newly added parts of the text. Generally, the manuscript is quite well-prepared and properly organized with the sufficient figures and tables, enough to be accepted for publication in this journal after minor revision.  I remain at your disposal for any clarification.

Author Response

Major revision taking into account the response point by point for the following:

Reviewer #1:

After the thorough check of the revised text, I have seen that the authors have implemented most of the proposed changes in the revised verion of manuscript towards the improvement of their work. Almost all the changes have been implemented and in my opinion, but a final check should be imeplemented towards the correction of some syntactical/grammatical errors in the newly added parts of the text. Generally, the manuscript is quite well-prepared and properly organized with the sufficient figures and tables, enough to be accepted for publication in this journal after minor revision.  I remain at your disposal for any clarification.

Thanks a lot for your advice. Based on your advice, the sentences in manuscript had been revised in revised manuscript.

Reviewer 2 Report

The work presented here has nothing to do with coatings.

I suggest to transfer the manuscript to another journal like molecules or Polymers where it is more appropriate.

Author Response

Reviewer #2:

I suggest to transfer the manuscript to another journal like molecules or Polymers where it is more appropriate.

Thanks a lot for your advice.

Reviewer 3 Report

I have rewritten some of the sentences that in my opinion are clearer now. Still, some paragraphs need to be written more clearly.

r.31: The density and raw material cost of the biomass wood-plastic composite material can be reduced by adding a foaming agent during the hot pressing process. At the same time, foaming can also imprint a unique texture to the surface of the product, improving its sound and heat insulation properties. Foamed wood-plastic composites were based on wood flour and plastic.”

r.68: “High-efficient flame retardant ammonium polyphosphate (APP) with molecular formula (NH4)n+2PnO3n+1 is a low cost, nontoxic and environmentally friendly inorganic phosphorus flame retardant compound.”

r.79-90: This paragraph must be rewritten. ZnO and Nano-CaCO3 were used as initiator and nucleating agent, respectively, for what reaction or process???? You must mention first the process that you want to initiate and nucleate using these compounds. Also you mention “the cell could prevent…”. About what cell are you talking about?

r.111-122: The paragraph remains very confusing. All those wt.% without mentioning relative to what. If these weight percentages are relative to the sum of the masses of PE-HD and Salix you should state that. Not that the total mass of the composite is the sum of the mass of PE-HD and Salix.

r.169: “This is because the polarity of APP/ZB and PE-HD matrix were different, which led to their poor compatibility.”

r.260: “The contact between oxygen and material can be diminished by the generated carbon layer.

Author Response

Major revision taking into account the response point by point for the following:

Reviewer #3:

  1. I have rewritten some of the sentences that in my opinion are clearer now. Still, some paragraphs need to be written more clearly. r. 31:“The density and raw material cost of the biomass wood-plastic composite material can be reduced by adding a foaming agent during the hot pressing process. At the same time, foaming can also imprint a unique texture to the surface of the product, improving its sound and heat insulation properties. Foamed wood-plastic composites were based on wood flour and plastic.”

Thanks a lot for your advice. Based on your advice, the sentence had been revised in revised manuscript. The details was as followed.

The density and raw material cost of the biomass wood-plastic composite material can be reduced by adding a foaming agent during the hot pressing process. At the same time, foaming can also imprint a unique texture to the surface of the product, improving its sound and heat insulation properties. Foamed wood-plastic composites were based on wood flour and plastic.

  1. r.68:“High-efficient flame retardant ammonium polyphosphate (APP) with molecular formula (NH4)n+2PnO3n+1 is a low cost, nontoxic and environmentally friendly inorganic phosphorus flame retardant compound.”

Thanks a lot for your advice. Based on your advice, the part had been revised in revised manuscript. The details was as followed.

High-efficient flame retardant ammonium polyphosphate (APP) with molecular formula (NH4)n+2PnO3n+1 is a low cost, nontoxic and environmentally friendly inorganic phosphorus flame retardant compound.

  1. r.79-90: This paragraph must be rewritten. ZnO and Nano-CaCO3 were used as initiator and nucleating agent, respectively, for what reaction or process???? You must mention first the process that you want to initiate and nucleate using these compounds. Also you mention “the cell could prevent…”. About what cell are you talking about?

Thanks a lot for your advice. This problem has been modified and the results are as follows.

The decomposition temperature of azodicarbonamide (AC) can be reduced by zinc oxide (ZnO). As a result, the foaming rate of AC is improved, and the foaming effect is better.

The addition of Nano-CaCO3 can increase the number of foaming sites and bubble of WPC, and make the bubble shape more uniform.

Therefore, foaming agent (AC), initiator (ZnO) and nucleating agent (Nano-CaCO3) were added to the Salix/PE-HD composites.

In revised manuscript, the “Cell” was revised as the “Cellular”. The foaming process can effectively reduce its density and weight. At the same time, Cellular structure can prevent crack propagation and improve the toughness of Salix/PE-HD materials. In addition, nano conductive carbon black (Nano-CB) and synergistic flame retardant APP/ZB were added to FWPC system to prepare flame retardant and antistatic Salix/PE-HD foamed composites. After APP/ZB is added to FWPC, it has better flame retardant and smoke suppression performance on the basis of maintaining its good mechanical properties. At the same time, its thermal stability is improved. Therefore, this study introduces a simple method to develop flame retardant, antistatic and high-performance FWPC, which has broad application prospects in the multi-functional utilization of wood and Packaging and transportation and decoration.

  1. r.111-122: The paragraph remains very confusing. All those wt.% without mentioning relative to what. If these weight percentages are relative to the sum of the masses of PE-HD and Salix you should state that. Not that the total mass of the composite is the sum of the mass of PE-HD and Salix.

Thanks a lot for your advice. This problem has been modified and the results are as follows.

The design density of FWPC was 0.9 g/cm3. The size of FWPC was designed to be 15×15×4 mm. The total mass of FWPC included the sum of PE-HD, Salix powder, Nano-CB and APP/ZB. The mass ratio of Salix wood powder and PE-HD was 4:6(m1:m2). As shown in Table 1, the fixed amount of coupling agent (KH550) was 3 wt%, the fixed amount of stearic acid was 0.6 wt%, the fixed amount of AC was 1.5 wt%, the fixed amount of Nano-CaCO3 was 3 wt% and the fixed amount of Nano-CB was 8 wt% (wt.% was the percentage of the total mass of FWPC). The fixed addition amount of zinc oxide was 20 wt% of the mass of foaming agent (AC). Composite ammonium polyphosphate (APP) and zinc borate (ZB) were added in mass proportion (4:1), and the amount of APP/ZB synergistic flame retardant was 0, 5 wt%, 10 wt%, 15 wt% and 20 wt%, respectively. The obtained FWPCs were named FWPC-0, FWPC-5, FWPC-10, FWPC-15, FWPC-20, respectively. (wt.% was the percentage of the total mass of FWPC. The following values of FWPC represent the amount of APP/ZB added).

Table 1. Formula of Flame Retardant and Antistatic Foamed Wood-plastic Composite.

Sample

APP/ZB

KH550

Stearic acid

AC

Nano-CaCO3

Nano-CB

FWPC-0

0

3

0.6

1.5

3

8

FWPC-5

5

3

0.6

1.5

3

8

FWPC-10

10

3

0.6

1.5

3

8

FWPC-15

15

3

0.6

1.5

3

8

FWPC-20

20

3

0.6

1.5

3

8

  1. r. 169: “This is because the polarity of APP/ZB and PE-HD matrix were different, which led to their poor compatibility.”

Thanks a lot for your advice. Based on your advice, the sentence had been revised in revised manuscript.

This is because the polarity of APP/ZB and PE-HD matrix were different, which led to their poor compatibility.

  1. r.260: “The contact between oxygen and material can be diminished by the generated carbon layer.

Thanks a lot for your advice. Based on your advice, the part had been revised in revised manuscript.

The contact between oxygen and material can be diminished by the generated carbon layer.